# Taxonomy-based approach for understanding and enhancing security culture in universities

Mona Albinali[1,2,3], Mahmood Niazi[1,2], Mohammad Alshayeb[1,2], Sajjad Mahmood[1,2] and Arif Ali Khan[4]

[1] Department of Information and Computer Science, King Fahad University of Petroleum and Minerals, Dhahran, Saudi Arabia
[2] Interdisciplinary Research Centre for Intelligent Secure Systems, King Fahad University of Petroleum and Minerals, Dhahran, Saudi Arabia
[3] Department of Computer Information Systems, Imam Abdulrahman Bin Faisal University, AL Khobar, Saudi Arabia
[4] M3S Empirical Software Engineering Research Unit, University of Oulu, Oulu, Finland

## ABSTRACT

**Context:** Recent studies have highlighted a growing interest in security culture. Frameworks and standards currently exist, offering cybersecurity regulatory guidelines for organizations both locally and internationally, as evidenced in the literature. However, research on information security culture (ISC) within universities remains limited. Moreover, there is a notable absence of professional and academic investigations into ISC.

**Objectives:** In this article we aim to thoroughly examine ISC in universities through four main objectives. First, we will identify essential knowledge areas (KAs) in this field. Second, we will analyze best practices (BPs) used to promote security culture in universities. Third, we will explore where these practices can be applied across different university areas. Finally, we will create a detailed taxonomy to organize the aspects of ISC in university settings.

**Method:** We employed a multivocal literature review (MLR) approach to identify the primary KAs and BPs for understanding and enhancing security culture in universities. We scrutinized 81 primary studies from formal (FL; peer-reviewed) and grey literature (GL; blogs, web pages, white papers). These studies span the past 14 years, from 2010 to 2024.

**Results:** We identified 12 main KAs and 76 best practice areas from both FL and GL. Our findings have enabled us to establish a taxonomy of security culture in universities. This comprehensive categorization serves as a valuable resource for understanding aspects of security culture.

**Conclusion:** This study will assist researchers and practitioners in finding relevant studies from the FL and GL to obtain evidence that will help develop an ISC model. Moreover, it has shed light on several areas that warrant further research and development to enhance security culture.



Corresponding author
Mona Albinali,
g201906770@kfupm.edu.sa

# INTRODUCTION

A primary issue in organizational management is information security, which has become a key strategy for information management. For most organizations, information security is a high priority because it is an organization's most valuable asset, and its security is crucial to economic stability. Thus, every organization should be concerned about the security of its assets (*Alkahtani, 2018*).

The increasing importance of information technology (IT), internet usage, and associated services has led to a rise in data attacks. Consequently, securing data has become essential. Information threats affect both public and private sector organizations, making data security a challenging role within information system management (*Al-Matouq et al., 2020*). According to information security surveys (*Department for Digital, Culture, Media & Sport; Ipsos MORI, 2020*), security threats are on the rise. Organizations are particularly vulnerable because employees may inadvertently open and click on attachments or links in phishing emails (*Department for Digital, Culture, Media & Sport; Ipsos MORI, 2020*). The financial repercussions of data breaches are significant, with organizations incurring costs related to data cleanup; data loss; liability; and a loss of customer loyalty, amounting to billions (*Alhogail & Mirza, 2014*).

One solution to enhance information security is to develop a culture of security awareness among employees, making them aware of the risks and their responsibilities. This approach encourages responsible and safe behavior, significantly reducing the risk of employee errors and inappropriate interactions with the organization's information assets (*Alhogail & Mirza, 2014*). Academics have suggested the development of a robust information security culture (ISC) as a means to ensure employees adhere to the organization's information security policy, especially in light of the increasing incidents and breaches caused by employee attitudes (*Nasir, 2019*). By fostering a strong organizational culture, information security can be effectively safeguarded (*Nasir, 2019*).

Information security encompasses various dimensions, including network security, data security, and cybersecurity. Within this framework, the ISC emerges as a critical dimension, underscoring the necessity for cultivating an ISC for successful management (*Nasir et al., 2019b*). There are numerous definitions of ISC. For example, *Alhogail & Mirza (2014)* described ISC as "the collection of perceptions, values, attitudes, knowledge, and assumptions that guide human interaction with information assets in an organization to influence employees' behavior in safeguarding information security." Similarly, ISC can be defined as the shared beliefs and values regarding information security across all organizational levels, incorporating social, cultural, and ethical approaches to foster appropriate behavior among members (*Malcolmson, 2009*). These perspectives highlight ISC as a culture that promotes safe behavior and aligns with the organization's objectives (*Nasir et al., 2019a*). An effective ISC includes guidelines for how individuals should behave while using IT systems, with the goal of avoiding actions that might threaten the security of data assets or IT resources systems. This culture is built on knowledge, artifacts, attitudes, and assumptions that more effectively promote security-related behaviors than mere regulatory demands on employee conduct (*Alhogail & Mirza, 2014*).

Organizations now prioritize information security, acknowledging that information is among their most valuable assets. Safeguarding these assets through strong security practices mitigates user behavioral risks, making it essential for economic stability. In this context, cultivating an ISC is vital. An ISC not only reduces risks but also transforms staff members into proactive defenders of data, often described as a "human firewall" (*Zakaria et al., 2007*). Recent studies have indicated that fostering an ISC can create a workplace culture where security is seen as "everyone's responsibility," which in turn improves compliance and encourages proper security practices at all levels (*Hogail, 2015*).

Nevertheless, the implementation and effectiveness of ISC differ widely among institutions, especially in educational environments. Universities encounter distinct information security challenges due to their varied data types, open network setups, broad academic collaborations, and specific data protection requirements. These issues are further complicated by sociocultural factors that shape security attitudes and behaviors within university communities. Differences in digital literacy and cultural expectations among students, faculty, and nonacademic staff can profoundly influence the effectiveness of security measures, creating specific vulnerabilities (*Morgan, Sibson & Jackson, 2022*).

It is crucial to address these sociocultural differences when designing security awareness programs that are culturally aware and inclusive. Such initiatives are vital for establishing an environment where security is a collective responsibility and protective behaviors are standard (*Mahmood, Chadhar & Firmin, 2024*). However, bridging these sociocultural nuances with robust information security practices remains underexplored in many academic settings. By incorporating strategies that account for diverse user behaviors and cultural expectations, universities can more effectively integrate secure practices. Moreover, according to the literature, some current frameworks and standards provide information security regulatory rules for organizations both locally and internationally (*Alnatheer, 2012*).

Nevertheless, work on an ISC focusing on universities is still lacking. Furthermore, during our literature review, we discovered professional and academic research on ISC in developing countries (*Alnatheer, 2012*). To the best of our knowledge, no study has focused on an explicit systematic literature review (SLR), especially a multivocal literature review (MLR), which offers comprehensive best practices (BPs) for implementing an ISC within universities, and no study has established a taxonomy of BPs for the ISC in universities. Accordingly, to bridge this gap, in this study we aim to conduct an MLR that offers a comprehensive analysis of the opinions of researchers and practitioners on security culture and establishes a taxonomy of BPs for security culture in universities. In this MLR we seek to deliver a comprehensive analysis of the current understanding in this field. By adopting this approach, we ensure unbiased research through careful evaluation of all available evidence and evaluating each study. Such methodology is crucial for generating accurate and reliable results (*Garousi, Felderer & Mäntylä, 2016*).

Unlike prior research by *da Veiga et al. (2020)* and *Tolah, Furnell & Papadaki (2021)*, which broadly addressed organizational ISC and key influencing factors, our study narrows its focus to the distinct context of universities and their unique operational elements (*e.g.*, teaching, research, administration). Through an MLR of 81 primary

sources, we identify 12 knowledge areas (KAs) and 76 BPs, culminating in a comprehensive taxonomy tailored explicitly to higher education institutions (HEIs). This dedicated attention to academic settings and the extensive inclusion of both formal literature (FL) and grey literature (GL) positions our work to fill a crucial gap, offering a more nuanced and practical framework for establishing and reinforcing ISC in universities.

The purpose of this MLR is to address the following research questions (RQs):

RQ1: What are the knowledge areas of the information security culture in universities?

RQ2: What are the best practices of the information security culture in universities?

RQ3: What are the domain applications to which the proposed information security culture solutions can be applied in universities?

RQ4: What is the taxonomy of the information security culture in universities?

The remainder of this article is organized as follows: In the Background section we summarize and outline the relevant research and background information. In the Literature Review section, we outline the literature review, and in the Methodology section we provide the research methodology employed to carry out the study. The results of the investigation are presented in the Multivocal Literature Review Results section and Demography Details of the Published Research section. In the Discussion section we examine the core findings of the study's outcomes. In Limitations and Implications, we provide detailed explanations of the limitations and implications of the research. The study is concluded in Conclusion and Future Work section, where we discuss potential future research.

# BACKGROUND

Before moving forward with the MLR, it is essential to provide background information on the importance of security in universities and the necessity of cultivating a robust security culture within the educational institution.

## Importance of information security in universities

Security awareness has been extensively studied in educational institutions over the years. Given the wide range of increasingly complex and aggressive attacks, it is critical to continually advise students and staff on adapting to technological changes, risks, and the requirements of new cyberspace (*Garba et al., 2021*). Moreover, with the digitalization of educational institutions, HEIs are more exposed to cyberattacks than any other type of institution. However, there is a notable lack of security awareness among HEIs, possibly because there are no specific standards or guidelines that can be universally applied or enforced. Additionally, most organizations do not monitor their employees' misbehavior (*Garba et al., 2021*).

*Parrish et al. (2018)* emphasized the necessity of information security education to protect against unauthorized use of electronic information. Educational institutions are identified as vulnerable and are regularly targeted by ransomware, leading to data breaches. Consequently, educational institutions must assess their systems and implement security strategies to enhance data security approaches and policies. For example, the University of Michigan experienced a data compromise, allowing unauthorized access to sensitive

information (*Parrish et al., 2018*). A lack of technical skills and social awareness of security issues heightens the risk of security vulnerabilities.

Therefore, HEIs must understand previous HEI hacking incidents, be prepared to protect against potential cyberattacks on their digital and PC systems, and enhance security at all levels. This preparation is crucial because cybersecurity experts have found that better cybersecurity education can prevent security attacks in other critical sectors, such as health care, emphasizing that education is fundamental to any professional career (*Almomani, Ahmed & Maglaras, 2021*).

Improving information security education and skills is one of the four critical components of the United Kingdom's 2011 national strategy to secure cyberspace (*Garba et al., 2021*). Starting from age 11, the United Kingdom's cyber strategy has incorporated information security at all education levels. The importance of information security has been underscored by an increase in cyberattacks in the education sector, which have become more common, resulting in financial losses and compromised data security. Preliminary findings from a UK information security survey in 2020 showed that educational institutions were among the most frequent targets in successfully reported data attacks (*Garba et al., 2021*). Therefore, highlighting the significance of information security in academic institutions as critical to helping others understand security training programs and protection methods (*Alhogail & Mirza, 2014*).

Because of the threats identified in the educational field, it is necessary to provide strategic recommendations based on practical security approaches. Additionally, management or educational authorities can assist in aligning security objectives with academic policies at individual institutions. According to *Da Veiga & Martins (2015)*, cybersecurity policies enhance risk management and guide the education sector in addressing cyber threats (*Garba et al., 2021*).

## Information security culture

According to *Schlienger & Teufel (2003)* and *Mahfuth et al. (2017)*, ICS is defined as a goal that must be achieved by fostering a culture that supports all related activities. Information security is becoming an integral part of every employee's daily responsibilities. *Ngo, Zhou & Warren (2005)*, and *Martins & Elofe (2002)* described ISC as the way individuals and the organization behave in a manner that naturally aligns with information security standards (*Alhogail & Mirza, 2014*). In research by *Martins & Elofe (2002)* and *Ngo, Zhou & Warren (2005)*, ISC is defined as the activities that the organization and employees should carry out in tandem to achieve compatibility and consistency with information security principles.

Creating and maintaining a security culture within an organization stems from the human aspect of information security, often regarded as the weakest link. Consequently, successful information security management requires establishing an ISC (*Alnatheer, 2012*). Therefore, a security culture is viewed as a subculture of an organization that encompasses employees' daily duties, activities, standards, and practices that must aid them in protecting information assets (*Mahfuth et al., 2017*). Furthermore, it is crucial to develop an ISC where workers can incorporate acceptable information security practices into their daily behavior for effective information security management. This implies that

employees should adopt security-aware behaviors in their work, a vital factor for managing information security (*Karlsson & Hedström, 2014*).

## LITERATURE REVIEW

Today, as the number of information security events rapidly increases and becomes more sophisticated and aggressive, the security of information systems has become a top priority. Any organization utilizing information technologies must regard information security as a critical priority (*Al-Janabi & Al-Shourbaji, 2016*). Organizations in developed countries suffer financial losses due to data breaches, security breaches, and security attacks. These incidents adversely affect an organization's reputation, financial standing, and commercial confidence. Moreover, the significance of information security management is escalating because of globalization and international competition (*Alnatheer, 2012*). One solution is to foster a security-aware culture that educates employees about risks and responsibilities, empowering them to respond safely and appropriately. Consequently, organizations should strive to foster a culture of information security so that employees have the essential knowledge and skills to behave appropriately and adopt a "secure" strategy (*Alhogail & Mirza, 2014*).

### Awareness levels in universities

A critical behavioral aspect of universities is the creation of an ISC. Despite the recognized need to develop a security culture to maintain and manage information security, it is widely acknowledged that organizations, particularly universities, face significant challenges in securing information systems (*Alkahtani, 2018*). When examining university awareness levels, a key theme surfaces: the differences in understanding of information security among students and staff. For example, *Senthilkumar & Easwaramoorthy (2017)* conducted a study among college students in Tamil Nadu, India, to assess their cybersecurity awareness. Their study found that 69.45% of students had a basic understanding of cybersecurity issues, which is considerably higher than average. As a result, Senthilkumar et al. recommend the implementation of higher-level security awareness and training programs to ensure students can protect their data from cyberattacks. However, a significant percentage of students were unaware (*Senthilkumar & Easwaramoorthy, 2017*).

Accordingly, the security of information systems is becoming more complex and challenging in knowledge-intensive organizations such as universities (*Alkahtani, 2018*). All universities must secure their students' data, eliminate risks, and minimize or mitigate any potential related risks. The lack of understanding among students and academic staff members about the threats and risks they may face in cyberspace can lead to the successful execution of such threats. Therefore, instilling a culture of information security awareness among students and academic staff members is crucial. Hence, measuring universities' information security awareness is considered one of the most critical aspects of this process (*Aljohni et al., 2021*). Moreover, several studies have addressed security issues, especially in educational environments. For example, *Al-Janabi & Al-Shourbaji (2016)* investigated data security awareness in the Middle East, specifically in educational institutions, including

researchers, undergraduate students, employees, and faculty members. The findings indicate a considerable lack of awareness of information security concepts among participants, who go about their daily work without the necessary knowledge of the importance of information security (IS) fundamentals.

## Behavioral aspects of information security

Behavioral aspects of ISC represent another key theme (*Sharifi, 2023*). Studies have shown that despite a certain degree of awareness, behaviors and practices concerning information security can differ significantly (*Sharifi, 2023*). Within the context of Saudi Arabia, a growing body of literature has recognized the importance of information security in Saudi Arabia's HEIs (*Masmali & Miah, 2023*; *Aljohni et al., 2021*). For example, *Aljohni et al. (2021)* aimed to assess the information security awareness level among university students in Saudi Arabia. They stated that securing the safety of cyber environments must be the top priority of every educational system (*Aljohni et al., 2021*). A survey conducted to collect students' awareness of cybersecurity threats in Kingdom of Saudi Arabia (KSA) universities showed that students should be taught security issues at a young age (*Aljohni et al., 2021*). The earlier they learn about the risks to data security, the better prepared they will be in the future because they will focus on security (*Aljohni et al., 2021*). The survey concluded that there is a lack of emphasis on essential concepts that must be included when assessing and analyzing university students' information security awareness in KSA (*Aljohni et al., 2021*). According to the findings, citizens of Saudi Arabia understand IT well but are unaware of information security practices and threats. The study suggested developing a framework to raise information security awareness among university students in the Middle East to prevent cyberattacks (*Aljohni et al., 2021*).

## Addressing unique security culture challenges

A focus on building security culture and practices has been increasing across various sectors, particularly within organizational and educational contexts (*da Veiga et al., 2020*; *Hina, Panneer Selvam & Lowry, 2019*). Studies have emphasized that security culture encompasses shared values, beliefs, and behaviors aimed at protecting organizational assets from security threats (*da Veiga et al., 2020*). These investigations span multiple domains, including general organizations, supply chains, health care, and HEIs, with varying methodologies and insights (*da Veiga et al., 2020*).

In organizational contexts, studies such as that by *Alhogail & Mirza (2014)* have provided foundational definitions and reviews of ISC, focusing on systemic values and norms. Similarly, *Karlsson, Åström & Karlsson (2014)* analyzed state-of-the-art research in ISC but noted a lack of recent advancements (after 2013). Studies such as those by *Lim et al. (2009)* and *Tang, Li & Zhang (2016)* have highlighted the interplay between organizational culture and security practices through qualitative methods, stressing the need for robust frameworks that integrate technical, managerial, and human-centric elements.

In the educational domain, there is a clear recognition of HEIs' unique vulnerabilities, such as decentralized IT structures and the need for awareness and governance strategies.

For example, *Kumar et al. (2024)* proposed a cybersecurity framework emphasizing governance, leadership, and risk assessment tailored to HEIs. Similarly, *Mahmood, Chadhar & Firmin (2024)* adopted a sociotechnical systems perspective to address security challenges in HEIs, focusing on organizational and crisis-specific interventions. These studies underline the need for practical, validated frameworks to enhance cybersecurity resilience.

Further research specifically within university settings includes that of *Rezgui & Marks (2008)*, who explored security awareness levels, and *Durojaiye, Mersinas & Watling (2020)*, who empirically investigated cybersecurity culture in UK HEIs. Although valuable, these studies have faced limitations such as narrow geographical focus or lack of empirical testing for proposed frameworks. Moreover, works such as those by *Furnell & Clarke (2005)* and *Kofi et al. (2020)* have explored conceptual and domain-specific security frameworks without addressing broader institutional challenges.

Considering the lack of research on the taxonomy of security culture in universities, in this research we aim to resolve this problem by providing a taxonomy of BPs for universities. Several researchers (*i.e.*, *Furnell & Clarke, 2005*, *Kofi et al., 2020*) have mentioned that their literature reviews reveal security concerns about universities. However, no explicit SLR, especially MLR, offers comprehensive BPs for implementing a taxonomy of ISC within these universities. To address this gap and provide an extensive examination of security practices relevant to universities, we conducted this MLR. The subsequent sections detail the MLR methodology employed in this study. An MLR approach holds greater value than an SLR in exploring security culture. A comparison of these reviews is presented in Tables 1 and 2. This comparison considers the proposed taxonomy, MLR methodology, and university context.

As shown in Tables 1 and 2, none of the previous studies provided a taxonomy of security BPs for universities using MLR. This motivated us to create such taxonomy through MLR.

By exploring the unique IS challenges that universities face and recognizing sociocultural factors that Impact security behaviors, in this study we aim to provide actionable insights for fostering a proactive, secure environment within educational settings, ultimately contributing to a framework that prioritizes information security in academia.

## RESEARCH METHODOLOGY

In this study we utilized an MLR, adhering to the guidelines set forth by *Garousi, Felderer & Mäntylä (2019)*. An MLR systematically incorporates FL and GL sources. FL includes journal articles that have undergone peer review as well as conference articles, whereas GL encompasses white articles, magazines, government reports, videos, and blog reports. The inclusion of GL in the review allows for the incorporation of a vast body of knowledge that is continually generated by industry practitioners outside of academic forums. An MLR provides significant value to both researchers and practitioners by offering evidence from both state-of-the-art research and the state of practice within a specific field (*Garousi, Felderer & Mäntylä, 2019*). Figure 1 illustrates the stages of the MLR conducted in this study.

**Table 1 Comparative analysis of studies on security culture.**

| Reference | Contribution | Method used | Limitation | Context | Type of review |
|---|---|---|---|---|---|
| Nnorom, Ezenwagu & Nwankwo (2020) | Explored modern security practices to enhance university administration | Case study approach | Focus on managerial perspectives, lacking actionable steps for technical security | University administration | Case study |
| Furnell & Clarke (2005) | Addressed security awareness and its integration within university culture | Conceptual framework | Did not test the conceptual framework empirically | Higher education institutions | Conceptual study |
| Kofi et al. (2020) | Introduced a nuclear security framework specific to educational settings | Qualitative analysis | Focused only on nuclear-related security aspects | Universities and academic institutions | Theoretical study |
| Durojaiye, Mersinas & Watling (2020) | Analyzed factors influencing cybersecurity culture in universities | Interviews and surveys | Focused only on specific UK institutions | UK higher education institutions | Empirical study |
| Rezgui & Marks (2008) | Studied the levels of security awareness in educational settings | Exploratory study | Did not address cultural differences across institutions | Higher education institutions | Exploratory research |

**Table 2 Checklist of studies on security culture.**

| Study focus/Feature | Nnorom, Ezenwagu & Nwankwo (2020) | Furnell & Clarke (2005) | Kofi et al. (2020) | Durojaiye, Mersinas & Watling (2020) | Rezgui & Marks (2008) | This article |
|---|---|---|---|---|---|---|
| Focus on security culture | ✓ | ✓ | ✓ | ✓ | ✓ | ✓ |
| University context | ✓ | ✓ | ✓ | ✓ | ✓ | ✓ |
| Build taxonomy from findings | | | | | | ✓ |
| Using the MLR methodology | | | | | | ✓ |
| Addressing cultural differences | | | | | | ✓ |
| Focus on technical controls | | | ✓ | | | ✓ |
| Integration of awareness and training | | | | ✓ | ✓ | ✓ |
| Holistic approach (governance + technical + culture) | | | ✓ | ✓ | | ✓ |
| Limitations clearly addressed | | | | | | ✓ |

## Phase 1 (Planning the multivocal literature review)

The first stage of the MLR involves the planning phase, which entails formulating research questions that encompass the review's primary objectives. Data are gathered from two sources, FL and GL, using the MLR approach. In the subsequent sections, we will cover both data extraction processes.

## Phase 2 (Conducting the multivocal literature review)

### Identify data sources

To ensure the collection of the most suitable data, we followed the recommendations of Zhang, Babar & Tell (2011) and White et al. (2001) by combining automated and snowballing methods. This approach allows for a comprehensive data collection process.

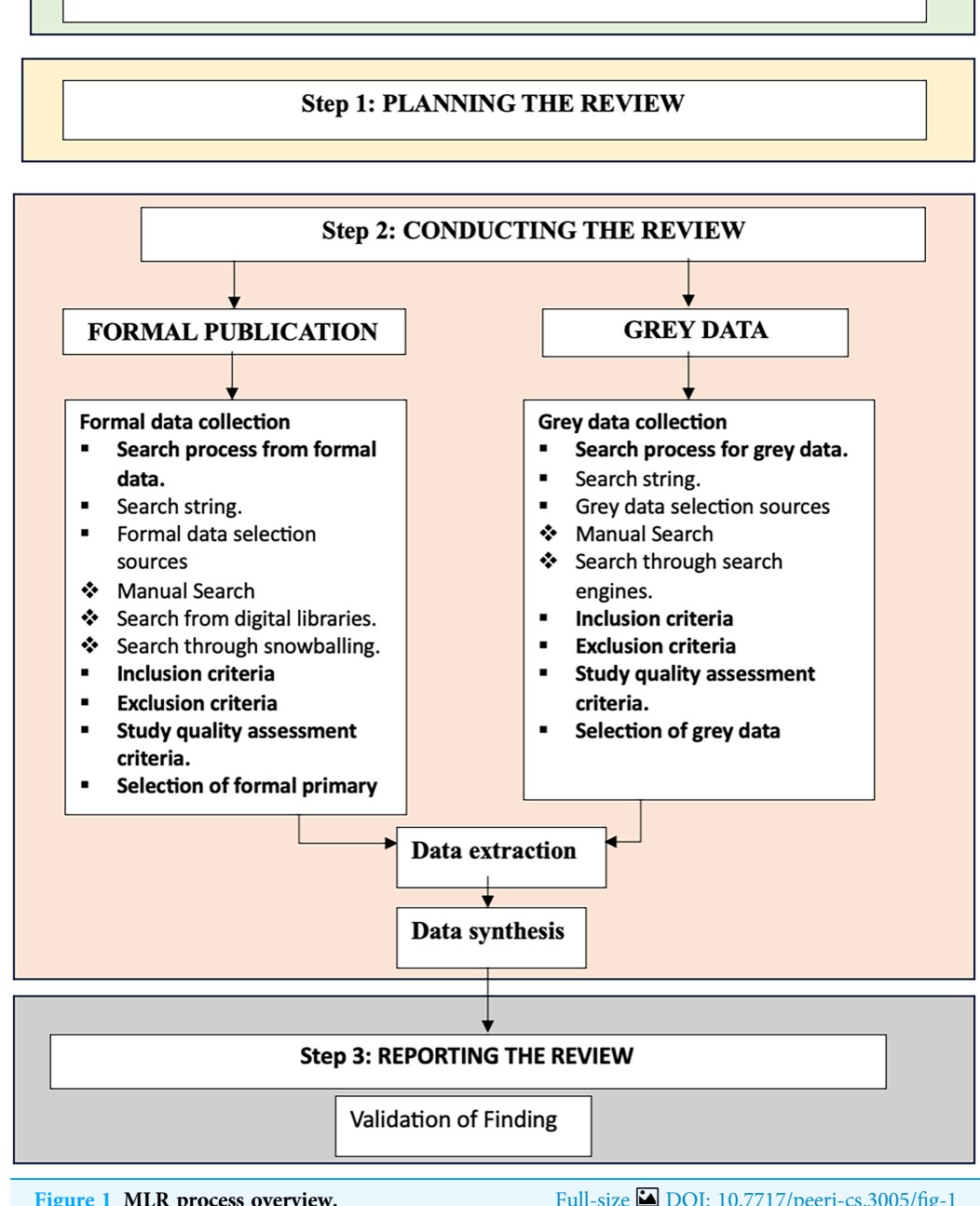

**Figure 1** MLR process overview.

- **Formal literature**

The academic digital libraries utilized in our research are listed in Table 3, along with their URLs. We conducted searches across four digital libraries, including IEEE Xplore, Scopus, Science Direct, and Springer, which are among the most frequently used sources for peer-reviewed literature on security culture. In the automated search, we utilized the

**Table 3 List of digital library sources.**

| Research sources | URL of advance search |
|---|---|
| Scopus | https://www.scopus.com/home.uri |
| Science direct | http://www.sciencedirect.com/science/search |
| IEEE Xplore | http://ieeexplore.ieee.org/search/advsearch.jsp |
| Springer | https://www.springer.com/gp |

advanced search feature to align the search string with the titles, abstracts, and keywords of articles published between 2010 and 2024.

- **Grey literature**

We conducted a manual search using Google Scholar to comprehensively understand the latest literature and verify the sufficiency of literature on the research topic for conducting an MLR. We utilized the search string "security culture" in the Google search engine. The recovered records fall into various categories, such as white articles, blogs, websites, news, and more. The Google search engine yielded many results matching our search query, but we limited the relevant results to the first 10 pages because they encompass the most essential information about our topic. Additionally, in this initial search we aimed to gather a collection of main research that may be utilized to validate the search term. We obtained 29 primary studies that are highly relevant to the research question.

- **Search through snowballing**

We have implemented a snowballing approach, which involves both backward and forward snowballing methods, using the reference lists of the selected primary studies from both phases (manual and automated search). The backward snowballing method involves manually examining the reference lists of selected primary studies to uncover any relevant research that might have been missed in the search string-based review. Using this approach, we discovered 10 more studies that clearly satisfy the inclusion and exclusion criteria.

### Formulate a search strategy

We developed a search string to facilitate a thorough exploration across numerous databases. To search the academic literature, we crafted a search string focusing on the following aspects: (a) keywords derived from the research questions, (b) synonyms and alternative phrases related to the identified terms, and (c) the combination of these terms using the logical operators AND or OR.

We restricted the search to the period from 2010 to 2024. We implemented this time constraint because of a significant increase in research focused on security culture compared to the number of studies published before 2010. By conducting multiple pilot searches, we meticulously refined our search criteria to ensure the inclusion of all relevant

primary studies. We also tailored the search query for various libraries. Ultimately, after several iterations, we identified the selected search string that we will utilize in this study:

(Information or "Cyber security") AND ("Security Culture" OR "Information Security Culture" OR "Cyber security Culture" OR "culture of information security") AND ("initiative" OR "method" OR "patterns" OR "practice" OR "activity" OR "approach" OR "process" OR "steps" OR "technique" OR "technology" OR "model" OR "framework" OR "guideline") OR ("university" OR "Higher Institutions"). (All Sources (Computer Science)).

The main objective of the final search string is to find the most relevant studies while minimizing the inclusion of potentially unrelated research, which could be time-consuming to review manually by examining titles, keywords, and abstracts for study selection.

### Define inclusion and exclusion criteria

We defined criteria for determining which articles to include and exclude. The selection criteria encompass both inclusion and exclusion criteria. Utilizing our inclusion and exclusion criteria ensures that only relevant sources are selected for subsequent analysis. Our inclusion criteria specified that articles must be in English, with full text available, and cover both FL and GL. Specifically, we sought articles with rigorous validation within educational institutions and those discussing challenges and practices related to security culture. To enhance our review, we expanded our GL to include a wider array of authoritative sources, such as white articles and governmental reports, ensuring comprehensive coverage. Conversely, we excluded non-English articles; those inaccessible in full; sources that narrowly focused on specific techniques without broader implications; and red, redundant publications. Tables 4 and 5 provide the inclusion and exclusion criteria for the literature review, including the corresponding codes (Inclusion 1–3 and Exclusion 1–5).

### Quality assessment

We assessed the selected articles for quality using criteria designed to eliminate research bias and measure the significance and completeness of the studies as described in *Garousi, Felderer & Mäntylä (2019)*. The sources included in the final review underwent a quality assessment (QA) to ensure their validity and avoid any bias. We evaluated the GL and FL sources according to the criteria outlined in Tables 6 and 7, respectively.

Based on the criteria adapted from *Garousi, Felderer & Mäntylä (2019)*, outlined in Tables 4 and 5, we evaluated both GL and FL. For GL, we assessed factors such as the publisher's reputation, clarity of objectives, transparency of methods, contributions to the field, and presentation of practices. For FL, our criteria included the foundation of research, clarity of goals, detailed methodology, empirical validation, discussion of limitations, practical applications, and discussion of findings. Each aspect was scored (Yes = 1, No = 0, Partly = 0.5) to include only high-quality, reliable, and relevant studies, enabling a thorough research analysis. articles that did not meet 50% of the quality criteria have been removed, ensuring a rigorous and focused research analysis.

**Table 4 Inclusion criteria.**

| Code | Inclusion criteria |
|---|---|
| Inc 1 | Articles in English and the full text are accessible (formal and grey literature). |
| Inc 2 | Articles include proper validation through a reliable methodology (formal literature). |
| Inc 3 | Articles report challenges and practices related to security culture on any organization including educational institution (formal and grey literature). |

**Table 5 Exclusion criteria.**

| Code | Exclusion criteria |
|---|---|
| Exc l | Studies that are irrelevant to our research questions |
| Exc 2 | Articles that focus on physical infrastructure or hardware |
| Exc 3 | Articles that focus on enhancing algorithms or features of a single security solution |
| Exc 4 | All duplicate articles found from various sources |
| Exc 5 | Articles not related to security culture in Universities |

**Table 6 Quality assessment criteria for grey literature adopted from *Garousi, Felderer & Mäntylä (2019)*.**

| | Criteria | Yes (1) | No (0) | Partly (0.5) |
|---|---|---|---|---|
| Q1 | Is the source published by a reputable organization? | | | |
| Q2 | Does the source have a clear goal? | | | |
| Q3 | Does the source have a clear methodology? | | | |
| Q4 | Does the source have a contribution? | | | |
| Q5 | Are the practices presented? | | | |
| Total | | | | |

**Table 7 Quality assessment criteria for formal literature adopted from *Garousi, Felderer & Mäntylä (2019)*.**

| | Criteria | Yes (1) | No (0) | Partly (0.5) |
|---|---|---|---|---|
| Q1 | Is the study based on research work? | | | |
| Q2 | Does the article have a clearly stated goal? | | | |
| Q3 | Does the article have a reported methodology? | | | |
| Q4 | Does the study include empirical evaluation? | | | |
| Q5 | Are the study limitations explicitly discussed? | | | |
| Q6 | Are the practices presented? | | | |
| Q7 | Are the research results discussed? | | | |
| Total | | | | |

## Source selection

Our methodology initially involved broad search criteria to capture a wide array of articles on security practices across different domains, including universities. We refined this through specific inclusion and exclusion criteria, focusing exclusively on articles discussing

ISC within universities. We further narrowed the selection by conducting a detailed QA of each article, ensuring relevance and depth in the context of ISC in higher education settings.

We selected the sources using a three-stage process based on the criteria previously mentioned by *Garousi, Felderer & Mäntylä (2019)*. In the initial phase, we gathered relevant sources by reviewing the titles and abstracts identified in the search results. In the second phase, we selected relevant sources by thoroughly examining the full texts of the initially chosen sources. During the final phase, we selected the sources that met the inclusion criteria and successfully passed the QA. According to the MLR standards, a source is chosen if it receives a score of 50% or higher in the quality evaluation for both types of assessments. Figure 1 displays certain research details obtained from various sources, following the inclusion and exclusion criteria. We obtained 52 scientific articles from a total of four distinct libraries as well as 29 grey studies. Appendix A presents detailed information on the FL.

### Perform data extraction

We retrieved relevant data from every chosen study to answer our four research questions. We exported the articles to a spreadsheet with columns for practices and additional categories, including the authors' names, publication country, venue (journal, conference, workshop, *etc.*), and year of release publication. These extracted data items are summarized in Table 8. We excluded a article if we identified during extraction that it did not fully meet the selection criteria.

### Perform data synthesis

The guidelines that (*Braun & Clarke, 2006*) presented for thematic analysis of qualitative data are considered a systematic approach to analyzing, organizing, and constructing themes based on the retrieved data. The core themes of retrieved data items are developed by following the steps of thematic data analysis:

**1. Familiarizing data:**

Researchers immerse themselves in the dataset by transcribing (if necessary) and repeatedly reading through the material. This stage involves noting initial impressions and ideas, which facilitates a deep understanding of the data's breadth and nuances.

**2. Creating initial codes:**

At this stage, the data are systematically examined to identify salient features. Researchers assign initial codes to meaningful data segments, creating a preliminary framework that captures key elements relevant to the research questions.

**3. Exploring themes:**

The next step involves collating the initial codes into potential themes. This process requires grouping related codes together, thereby uncovering broader patterns that reflect underlying concepts or issues across the dataset.

**Table 8** The extracted relevant data items from the chosen primary studies.

| Code | Data item | Description | Related RQ |
|------|-----------|-------------|------------|
| C1 | Index | The study ID | Demographic |
| C2 | Title | Full title of primary study | Demographic |
| C3 | Name of authors | Authors full names | Demographic |
| C4 | Publication year | Temporal information of each study | Demographic |
| C5 | Publication type | Journal, conference, workshop, book chapter, magazine | Demographic |
| C6 | What are the different knowledge areas of the information security culture? (factors-theme-dimensions-pillars) | Knowledge areas of the information security culture | RQ1 |
| C7 | What are the best practices in the article? (need to gather the best practices to build the knowledge areas) | Best practice for the information security culture | RQ2 |
| C8 | Type of articles (opinion-based, evaluation) | (Opinion-based, evaluation article, *etc.*) | RQ3, RQ4 |

**4. Analyzing themes:**

The emerging themes are rigorously evaluated against the coded data and the entire dataset to ensure coherence and distinctiveness. This review process helps to refine the themes, ensuring that they accurately represent the data and do not overlap in meaning.

**5. Defining and designating themes:**

Each refined theme is then clearly defined and assigned a descriptive label. This stage involves articulating what each theme captures and clarifying its significance within the overall analysis, thus enhancing interpretative clarity.

**6. Generating the report:**

The final step culminates in producing a comprehensive report. Here, the themes are integrated into a coherent narrative, supported by illustrative data extracts, and contextualized within the broader research framework and literature, ensuring that the analysis effectively addresses the research objectives.

## Phase 3 (Reporting the multivocal literature review)

We reviewed, assessed, gathered information, and summarized findings from the chosen articles based on the specified research questions outlined in the introduction. From each selected source, we extracted relevant data to respond to these research questions using a set extraction form.

### Quality assessment of primary selected studies

We conducted an assessment to evaluate the quality of the chosen data sources, including primary studies and GL, to determine their effectiveness. We utilized the QA criteria outlined in Tables 6 and 7 to evaluate the chosen data sets. Articles that did not meet 50% of the quality criteria have been removed, ensuring a rigorous and focused research analysis.

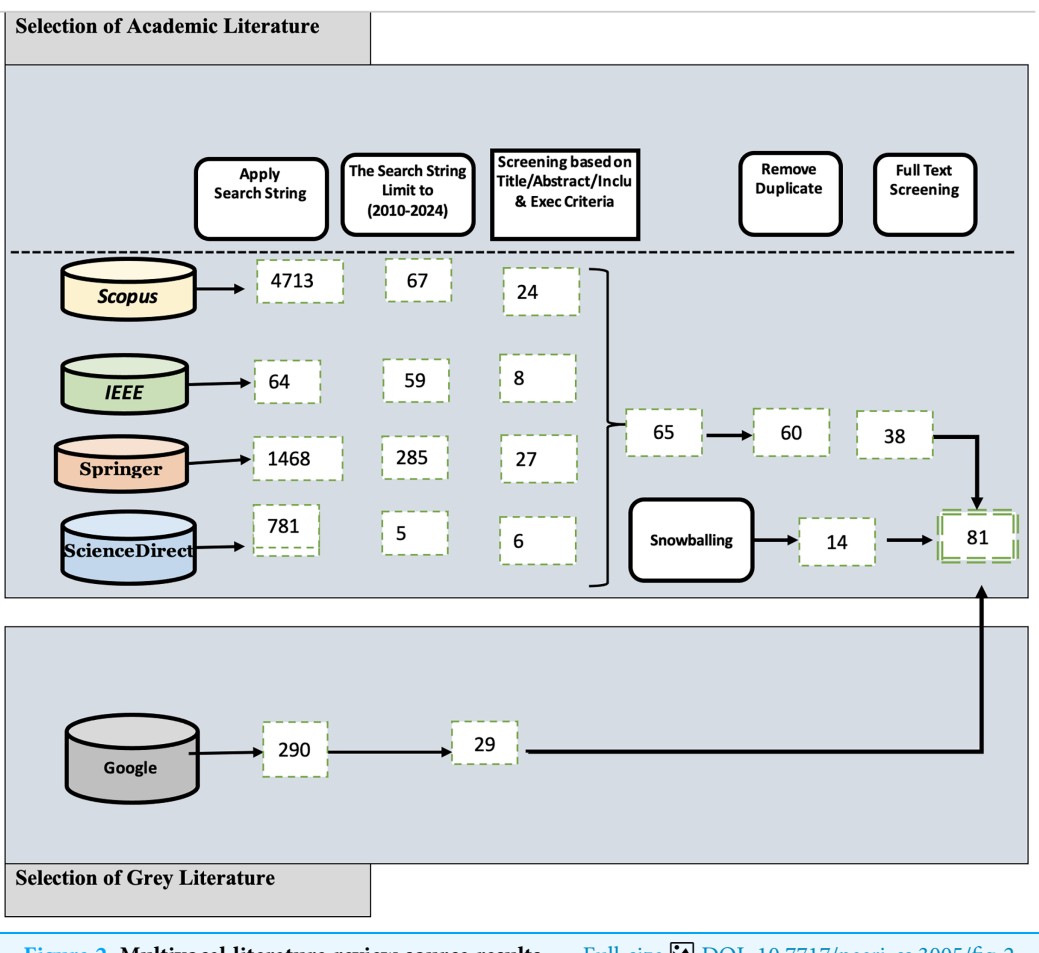

**Figure 2  Multivocal literature review source results.**

# MULTIVOCAL LITERATURE REVIEW RESULTS

As mentioned in the Research Methodology section, we obtained 416 sources from various digital libraries. Following this review, 65 sources successfully passed the second review stage, and ultimately, 38 passed the final review. Furthermore, we employed a backward snowballing method to manually examine the reference lists of the selected 14 primary studies. With this method we aimed to find additional studies that may not have been included in the search string review. Figure 2 provides a comprehensive overview of the selection process for both GL and FL at each stage of the MLR. The figure includes details such as the search databases and the corresponding number of articles selected at each respective step.

## Security culture knowledge area

In this section, we address RQ1. We employed thematic analysis to identify 12 KAs from the selected 81 studies using the MLR approach. Additionally, we utilized a bottom-up approach to uncover more KAs by grouping the BPs. All the identified KAs, along with their definitions, are presented in Table 9.

**Table 9 The extracted knowledge area with definitions.**

| No. | Knowledge area | Definition |
|---|---|---|
| 1 | Top management support, leadership, or involvement | The involvement and commitment of top-level executives, senior managers, and department managers in fostering and implementing a security culture |
| 2 | Security policy | A collection of guidelines and procedures that an organization establishes regarding security |
| 3 | Security awareness | The level of understanding of security among employees inside an organization |
| 4 | Security training | The process of providing education, instruction, and practical guidance to individuals within an organization to enhance their understanding and skills related to security measures, protocols, and best practices |
| 5 | Change management | The process of providing guidance and assistance to employees to foster the change for developing a security culture inside an organization |
| 6 | Compliance | A planned approach to ensuring that both employees and the organization as a whole strictly conform to established rules and regulations regarding security |
| 7 | Knowledge of security | Workers understand security hygiene practices as well as an organization's rules and procedures |
| 8 | Security risk analysis and assessment | The process of assessing and evaluating potential threats and vulnerabilities within an organization's systems, processes, and environment |
| 9 | Communication | The methods employed for sharing information between the organization and employees, such as the channels *via* which employees are informed about security policies, practices, and their expected responsibilities |
| 10 | Trust | Employees and the organization must have a level of mutual confidence, both in general and in relation to security activities; this mutual confidence extends to all operations within or related to the organization |
| 11 | Norms—ethical conduct | Behavioral and decision-making within an organization adhere to a moral code that distinguishes between what is right and what is wrong with a security culture |
| 12 | Attitude (behavior) | The beliefs, values, and actions that individuals within an organization demonstrate toward security practices and protocols |

## Best practices for security culture

In this section, we address RQ2 regarding the BPs for security culture. An important stage of our research methodology involved extracting data from the MLR, where we collected all raw practices derived from the MLR findings. We conducted a comprehensive analysis of primary studies to identify BPs for security culture. These identified BPs, derived from both FL and GL, are outlined in Appendix B. The numbers represent the 12 main KAs according to the ISC taxonomy. Each number is linked to a specific knowledge area, such as "1" for the "security policy" KA. Beneath each KA, there are sub-KAs (*e.g.*, 1.4 Account) that further detail the main KA. Each subcategory then breaks down into detailed security BPs (*e.g.*, in 1.4.1, a system will automatically log out after 10 min). This structured numbering approach organizes security practices clearly and shows their relevance to the broader ISC KAs, making the guidelines easy to follow and apply. Also, we have removed the color coding. Moreover, we employed snowballing techniques to enrich the findings further, enabling the discovery of additional practices from the literature. As mentioned before, we identified a total of 12 KAs, and under each KA, we identified a distinct set of BPs and linked them accordingly. In the subsequent stage, we undertook a careful consolidation process, merging certain practices while simultaneously eliminating redundant, ambiguous, or vague practices. This stage underwent a thorough review by the advisor to ensure the accuracy and coherence of the refined practices. The refinement

**Table 10  Review of the practices.**

**First review**

Action: Collecting all practices through a multivocal literature review

| | |
|---|---|
| **No. of practices:** | **190** |

**Second review**

Action: Removing redundant, ambiguous, and vague practices

| | |
|---|---|
| **No. of practices:** | **106** |

**Third review**

Action: Applying Motorola measurement to validate the practices

| | |
|---|---|
| **No. of practices:** | 76 |

process went through three iterations, as depicted in Table 10, which clearly represents the improvements made to the BPs.

## Theoretical validation

Once we finalized the BPs, we conducted a pilot study to obtain expert opinions on the strength and measurability of these practices. We selected three IT experts from University A based on their expertise in security rather than software development. Furthermore, the experts were required to have at least three to four years of relevant experience. We then asked them to apply the identified practices at their respective universities to assess their measurability. Based on the feedback received, we deemed certain practices inadequate and subsequently removed them, whereas we modified others based on the experts' comments. This iterative process allowed for the refinement of the practices.

We conducted the validation of the practices on two levels. First, we sought expert opinions, serving as the initial level of validation. Then, we asked the experts to assign scores to each practice, and we calculated an average score for each. To enhance clarity and measurability in practices, the Motorola assessment tool served as a theoretical validation method for the results from the second stage. This tool is a standard for measuring and verifying the implementation of specific practice activities. It has been extensively utilized to assess practices for each proposed model in multiple published studies (*Mufti et al., 2018*; *Khan, Niazi & Ahmad, 2009*; *Ali & Khan, 2016*). There are three strong reasons for utilizing Motorola's instrument: it is a normative tool that allows for adaptation; it has been previously tested successfully at Motorola, and it features a limited number of activities. Additionally, we used a theoretical validation approach by employing the opinions of two academic experts and the Motorola measurement tool to validate the results obtained in the second stage.

Following the validation process with the three IT experts and their valuable feedback, we developed a taxonomy of security culture in universities based on the experts' opinions. Through thematic analysis, this review facilitated a structured comparison and evaluation of current security culture solutions, as outlined in the Taxonomy of Information Security Culture in Universities section.

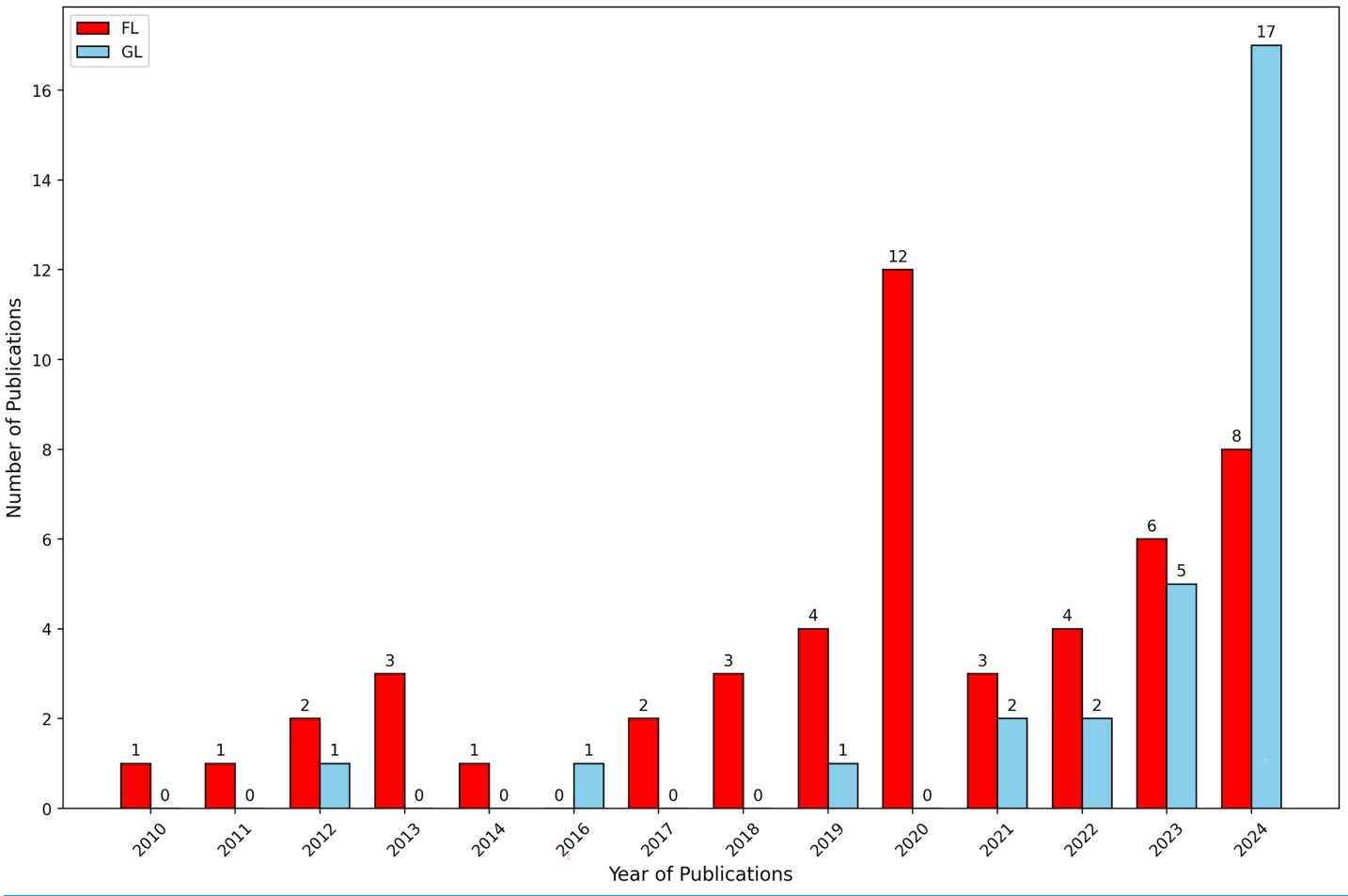

**Figure 3** **Annual number of sources in formal literature and grey literature.**

# DEMOGRAPHY DETAILS OF PUBLISHED RESEARCH

Within the MLR, we conducted a systematic mapping to examine the demographic characteristics of published research. With this mapping we aimed to explore various aspects, including the types and frequency of publications, the contributions of research studies, and the application domains in which these studies were conducted. In the following section we detail the findings obtained through this systematic mapping analysis.

## Types and frequency of publications

We conducted a data-driven analysis to examine the frequency of data publication in FL and GL from 2010 to 2024. With this analysis we aimed to assess the occurrence of publications over the specified period. The findings, depicted in Fig. 3, indicate that the frequency of FL has been consistently much higher than that of GL throughout the last 13 years examined.

Figure 4 displays the details of scientific literature sourced from four libraries. Based on the findings, it is evident that a significant portion of the studies addressing security culture issues and BPs are published in Springer. The results indicate that Springer has the highest

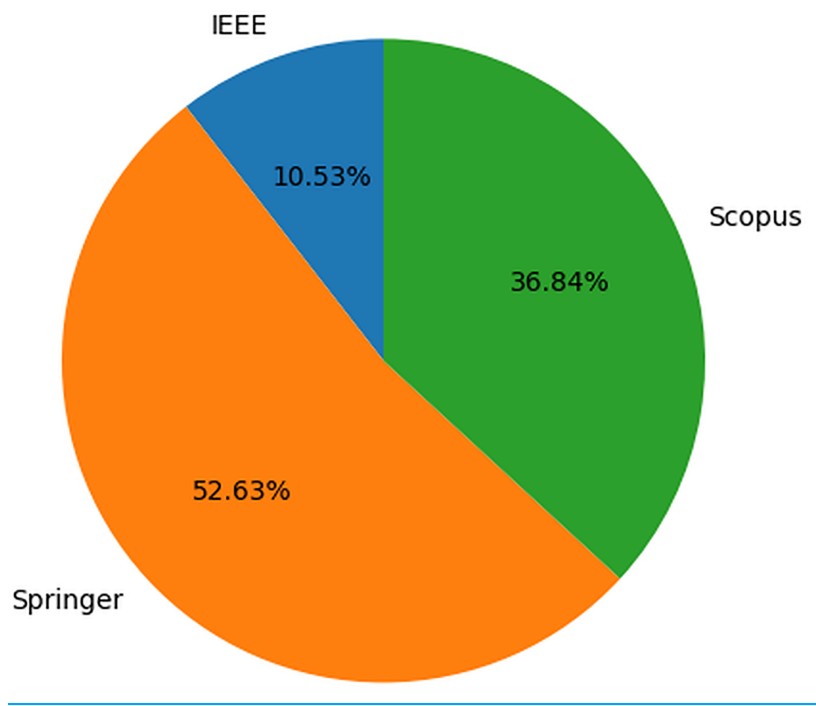

Figure 4 **Details of literature retrieved.**         

number of publications in this area compared to other sources, with IEEE having the lowest number of publications.

## Classification of application domains

In this section, we address RQ3 concerning the application domain for security culture. We utilized the thematic process outlined in "Research Methodology" to categorize the selected primary studies based on their common application domains. The systematic process of identification, categorization, and naming of themes and subthemes is illustrated in Fig. 5. Through this process, we were able to identify at least two or more studies that shared common application domains, which we then grouped under a single umbrella term: "theme." We have identified five core themes in this study, with the selected studies classified accordingly: (i) Information Security: This theme includes 89.47% of the total; (ii) Nuclear: This theme represents 2.63%; (iii) Intelligent Systems: This theme includes represents 2.63%; (iv) Information Systems and Technology: This theme represents 2.63% (v) Engineering: This theme represents 2.63%. These themes provide a comprehensive framework for organizing and analyzing the selected studies based on their respective application domains.

## Types of research and contributions

We have categorized the selected publications into well-established research types, as *Khan et al. (2023)*. These types include evaluation research, proposed solutions, validation research, philosophical essays, opinion pieces, and personal experience articles. Evaluation research focuses on analyzing a particular problem or solution in practice through diverse

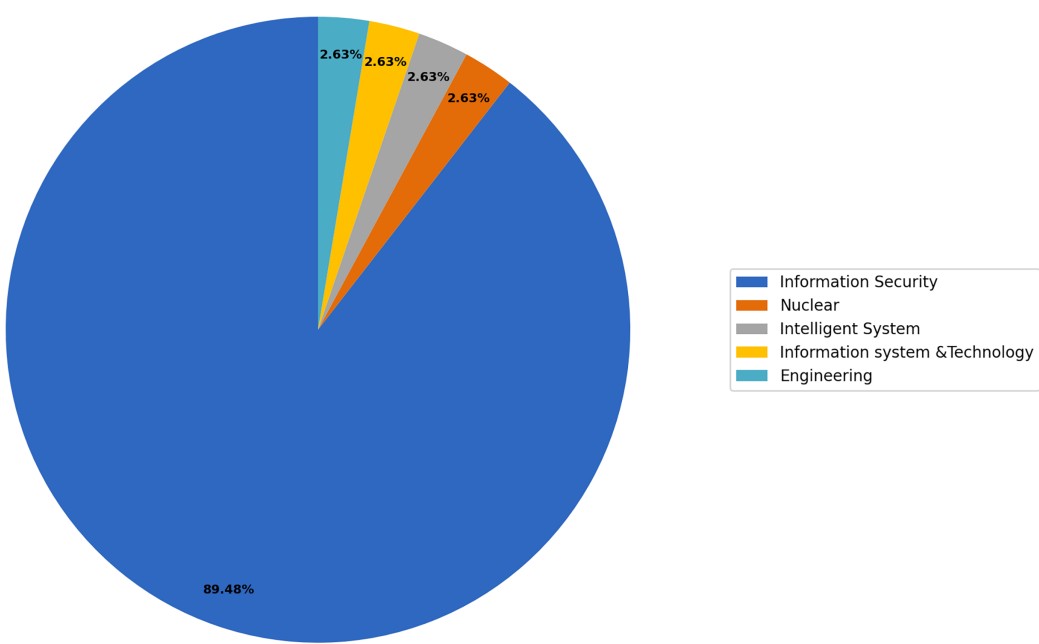

**Figure 5 Application domains-based thematic classification.**

empirical research methods. Articles proposing a solution focus on developing methods or solutions for relevant problems, although their significance may still need validation. Validation research aims to evaluate the quality attributes of proposed solutions that have yet to be implemented in real-world environments. Philosophical articles revolve around the establishment of theoretical or conceptual frameworks. Opinion articles express the authors' positive or negative views on specific frameworks, models, or solutions. Personal experience articles involve authors sharing their individual experiences related to specific projects or endeavors.

Furthermore, we have reported the research contributions of each article categorized within these research types. We employed the thematic analysis process, as discussed in "Research Methodology", to classify the selected 81 primary studies into the aforementioned research types. The set of selected studies consists of 10% solution-based empirical evidence from interview articles, 42% solution-based empirical evidence from questionnaires and surveys, 37% solution-based nonempirical evidence (opinion articles), and 11% personal experience/philosophical articles. We further classified these studies into a separate category, namely the proposal of solution and validation research, as depicted in Fig. 6.

## The taxonomy of information security culture in universities

In this section, we answer RQ4 regarding the taxonomy of the information security culture in universities. We created a taxonomy of security culture in universities based on the experts' opinions after the validation approach with the three IT experts and their valuable feedback.

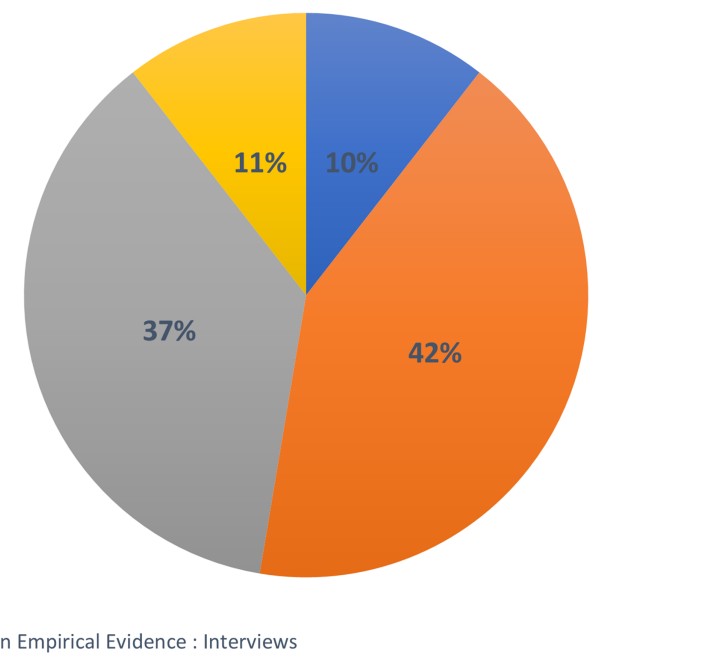

The solution based on Empirical Evidence : Interviews
The solution based on Empirical Evidence : Questionnaires and Surveys
The solution based on Non-Empirical Evidence (Opinion Papers) : Theory-based research(e.g., literature reviews)
Personal Experience/Philosophical Papers : Other(s)

**Figure 6  Type of published research.**

As shown in Fig. 7, the proposed taxonomy encompasses various dimensions and subdimensions to classify security culture KAs and practices. We have shown the 12 KAs in detail, and we have identified the BPs under each KA. Figure 7 shows a visual representation of the 12 KAs and BPs. These KAs are security policy, security awareness, security training, top management support, knowledge of security, trust, risk analysis and assessment, security compliance, norms and ethical conduct, communications, change management, and attitude.

## DISCUSSION

During this study, we identified KAs and BPs contributing to the development and improvement of ISC. We explored the important KAs for building and maintaining an ISC.

### Top management support

From examining the literature, it is evident that top management support is one of the most critical KAs identified in the MLR regarding security culture. The support and commitment of top management play a crucial role in fostering a strong security culture within an organization, including universities. This KA is essential because it sets the tone for the entire organization and influences the attitudes and behaviors of employees and students toward security. The importance of top management support in establishing and maintaining a strong security culture has been extensively documented, demonstrating

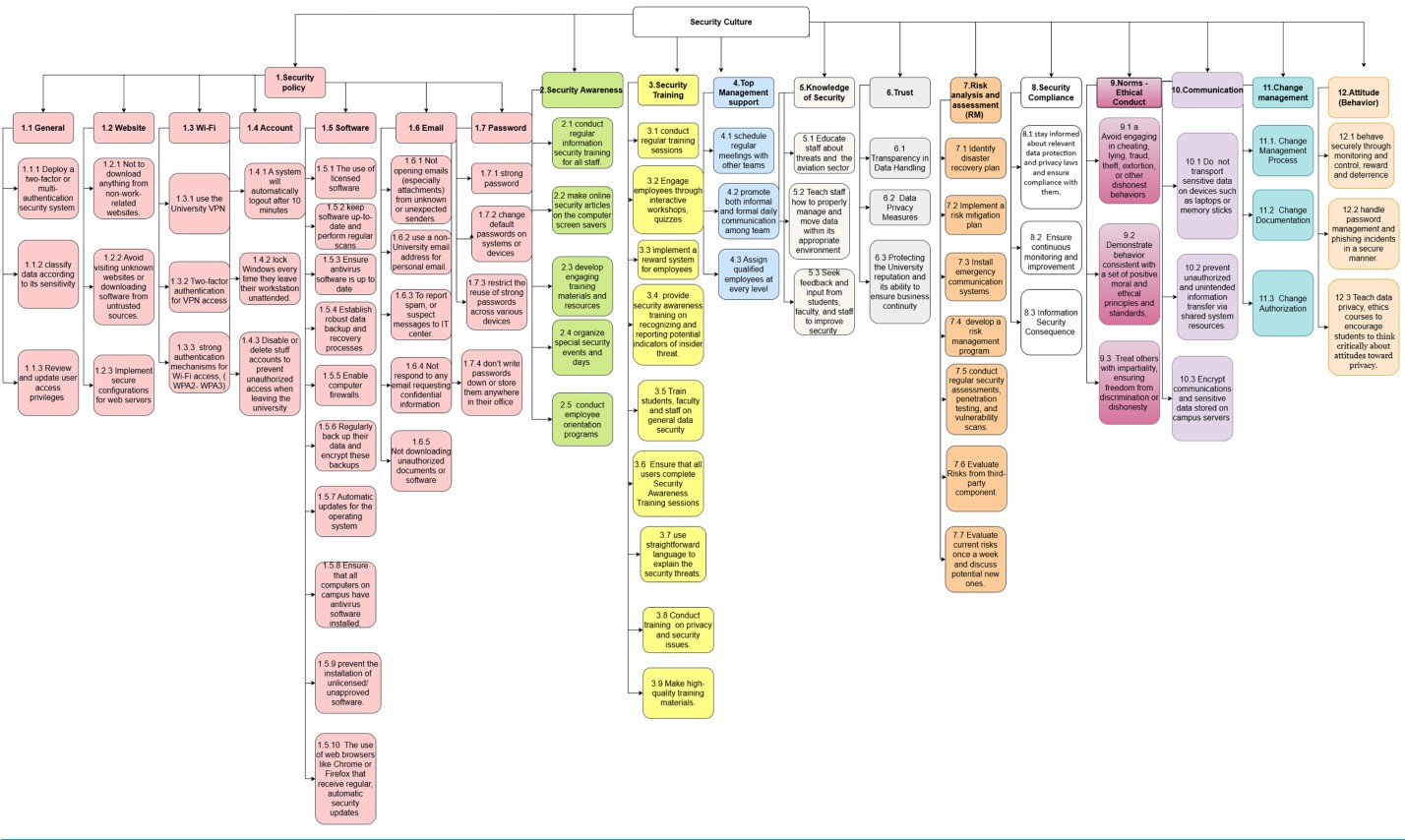

**Figure 7** The taxonomy of information security culture in universities.

that leadership commitment not only sets the tone for organizational behavior but also significantly influences employee and student attitudes toward security (*Hu et al., 2012*).

## Policy-driven security culture

**Security policies and procedures**: Another significant KA identified in the study regarding security culture is establishing and implementing effective security policies and procedures. These serve as the foundation for developing a robust security culture within universities and play a crucial role in guiding employees and students toward secure behaviors and practices. The analysis indicates that well-defined security policies and procedures are essential for establishing a common understanding of security requirements and expectations. These policies and procedures outline the rules, responsibilities, and consequences related to security practices, providing a framework for individuals to navigate the complex landscape of cybersecurity threats and risks. By implementing comprehensive security policies and procedures, universities can effectively address security challenges, such as data protection, access controls, incident reporting, and password management. Clear guidelines and procedures help individuals understand how to handle sensitive information securely, access systems and resources, and respond to security incidents promptly and appropriately. Research by *Hina & Dominic (2020)*

indicated that clearly articulated security policies considerably boost adherence among university staff and students.

**Compliance:** We identified compliance as an important KA in universities' security cultures. Compliance involves adhering to security policies, protocols, and regulations to protect sensitive information and resources. Promoting and enforcing compliance can help universities mitigate security risks; enhance data protection; and create a secure environment for students, staff, and stakeholders. The research conducted by *Hina & Dominic (2020)* demonstrated that robust compliance significantly strengthens the overall security framework by mitigating risks and highlights the essential function of enforcing security policies within HEIs, illustrating a correlation between stringent compliance, a reduction in security breaches, and an increase in awareness among faculty and students.

**Ethical behavior:** Ethical conduct and behavior are fundamental aspects of a strong university security culture, involving respecting privacy, adhering to legal requirements, and handling sensitive information carefully. Promoting ethical behavior through clear policies, training on ethical conduct, and enforcing consequences for misconduct contributes to a culture of security where individuals understand the impact of their actions and make conscious efforts to protect the university's assets and information. Maintaining ethical behavior, supported by clear policies and strict compliance, is critical for reinforcing a security culture. Ethical management of data and respect for privacy are crucial. *Guerrero-Dib, Portales & Heredia-Escorza (2020)* discussed the consequences of ethical breaches within educational environments and highlighted how ethical breaches can damage trust and reputation, potentially leading to legal consequences and weakening the overall security posture in educational settings.

## Change management in security practices

Effective change management is also identified as an important KA. It is significant in successfully implementing and embedding security practices into the organizational culture. Change management processes offer a structured approach to managing changes, ensuring they are implemented smoothly and effectively. By considering change management factors, universities can effectively overcome resistance to change. This approach enhances buy-in from stakeholders and facilitates the successful adoption of security practices. This is particularly important when universities undergo organizational changes, such as mergers or adopting new technologies, necessitating effective communication of the importance of security, alignment of security objectives with new organizational goals, and seamless integration of security into the new structure. *May & Stahl (2017)* argued that well-structured change management procedures are necessary to mitigate resistance and secure stakeholder support, particularly during technological upgrades or structural changes.

## Awareness and training programs

The MLR highlighted the significance of awareness and training programs as essential KAs. These programs aim to increase individuals' understanding of security risks, BPs, and

their roles and responsibilities in maintaining a secure environment. The importance of awareness and training cannot be overstated. These efforts are crucial for elevating understanding of security risks and BPs. *Adeshola & Oluwajana (2024)* emphasized that customized training programs significantly elevate security awareness throughout university campuses. Furthermore, research on differences in security culture by profession has indicated the need to customize awareness programs based on specific roles and responsibilities within the university (*Ramachandran et al., 2013*).

By enhancing awareness and providing training, organizations can empower their employees and stakeholders to make informed decisions and actively contribute to the security culture.

### Knowledge of security

Security knowledge is important in universities' development of robust security cultures. This knowledge involves providing education, training, and awareness programs to students, faculty, and staff about security threats, BPs, and preventive measures. Enhancing security knowledge empowers individuals to identify potential risks, make informed decisions, and take proactive steps to safeguard their digital and physical assets, contributing to a collective effort to maintain a secure environment (*Guaña-Moya et al., 2024*).

### Risk management

Conducting regular security risk analysis and assessment is crucial for finding potential vulnerabilities and threats, prioritizing security efforts, effectively allocating resources, and implementing appropriate security controls. Risk assessment also evaluates the effectiveness of security measures and identifies areas for improvement, helping universities minimize the likelihood of security breaches and protect their critical infrastructure and data. *Ariff et al. (2014)* demonstrated how continuous risk assessments can aid universities in adjusting their security strategies to protect critical infrastructure and minimize breaches.

### Communication

Effective communication is a key KA that fosters a strong security culture within universities. This communication involves clearly and timely disseminating security policies, incident reporting procedures, and educational materials. Establishing communication channels promotes awareness, shares security updates, and provides guidance on security-related matters. This transparency encourages individuals to report suspicious activities, seek guidance, and stay informed about emerging security threats. *Cheng et al. (2022)* discussed how establishing clear communication channels has successfully enhanced the responsiveness to security updates and incidents in educational settings, improving overall security posture.

### Building trust

Trust is another KA found in the present study, which is crucial for a successful security culture in universities. Trust is established when individuals have confidence in the

security measures that the university has implemented and believe their personal information and assets are protected. Demonstrating a commitment to security by implementing robust security controls, safeguarding sensitive data, and maintaining transparency fosters a sense of responsibility, accountability, and collective ownership in a secure environment. *Almadhoun, Dominic & Lai (2014)* explored how transparency and consistent enforcement of security policies build trust among university stakeholders, leading to more proactive participation in security initiatives.

The analysis offers important insights for both practitioners and researchers in information security at universities, offering practical guidance for building and managing a robust security culture. By addressing these KAs, universities can cultivate a strong security culture that fosters a proactive and resilient approach to information security. The findings also highlight the need for further research on topics such as change management processes and the influence of national security culture on security culture within universities.

## LIMITATIONS

We followed a systematic approach for this study's MLR, carefully evaluating sources based on defined quality criteria. However, it is important to acknowledge the potential limitations of this study. One limitation is the possibility of an incomplete literature review due to reliance on specific search keywords, limited digital databases, and search engines. To address this limitation, we made efforts to identify synonyms and similar spellings of search keywords, resulting in a robust search string. Additionally, we utilized multiple digital databases and search engines to explore a broad range of relevant literature. Another potential limitation concerns the researchers' biases in identifying and mapping practices into KAs. To mitigate this limitation, two other internal and one external researcher assisted in conducting an iterative selection process and comprehensive data extraction. We took an additional step to perform tests to examine and minimize any potential biases.

Furthermore, it should be noted that the review process was limited to studies published in English and those with full-text availability online. This selection criterion may have excluded valuable studies written in other languages or studies not freely accessible online. To address this limitation, we made efforts to include GL, such as reports, working articles, government documents, and white articles, that do not rely solely on peer-reviewed articles. Although we acknowledge these limitations, we mitigated them to the best extent possible within the scope of this study. Nonetheless, future research should consider expanding the language and accessibility criteria to ensure more comprehensive coverage of relevant studies.

## IMPLICATIONS

The objective of this study is to build a taxonomy of BPs for security culture in universities. We conducted an MLR to identify the main knowledge areas and BPs for addressing university security culture. We believe that the findings of this study are expected to greatly benefit both the research and industry communities by providing a better understanding of

security culture in universities. These findings will be valuable input for developing a security culture readiness model. By offering a list of BPs and a comprehensive body of knowledge, this study assists academia and industry experts in creating a readiness model that allows universities to assess their level of preparedness. Our future objective is to develop a readiness model for security culture that enables universities to measure their readiness level and implement the necessary BPs to foster an ISC. This model will provide guidelines, highlight strengths and weaknesses, and assist universities in enhancing their security culture. The development of this model will be based on evidence gathered from both the research and practitioner communities through this MLR.

## CONCLUSION AND FUTURE WORK

In recent years, there has been increased focus on studying security culture within organizations, particularly universities, recognizing the essential role of information security in safeguarding data, information systems, and assets against cyber threats. However, the critical need for a strong security culture is frequently neglected in organizational security strategies. In this study we identify the main KAs and BPs vital for fostering a security culture in universities. We have adopted MLR as it offers substantial value to both researchers and practitioners by presenting evidence drawn from cutting-edge academic research as well as real-world industry practices within a specific domain. Through a thorough MLR, we examined 81 studies, including 52 FL and 29 GL studies, sourced from four major databases spanning from 2010 to 2024. We introduced a taxonomy for security culture within universities, which we further divided into 12 KAs and 76 BPs, offering a structured perspective on this critical domain. To our knowledge, this is the first MLR on university security culture that offers comprehensive BPs for implementing a taxonomy of ISC within universities. Despite these advances, the realm of ISC in higher education remains underexplored and requires more focused scholarly investigation. Future studies should target three key areas: First, developing a "security culture readiness model" specifically designed for HEIs, which includes defining essential performance indicators for security practices and testing the model in various environments to ensure its adaptability and efficacy. Also, it is essential to empirically validate the proposed taxonomy across various university settings to confirm their effectiveness and reveal areas for further improvement, ensuring these models can effectively foster a proactive and resilient security culture. By addressing these identified gaps and improving methodological transparency, future research can strengthen theoretical models and enhance the practical application of security measures within universities, allowing institutions to proactively mitigate cybersecurity risks while aligning with academic values and standards.

### Funding

This research was supported by the Interdisciplinary Research Centre for Intelligent Secure Systems, King Fahd University of Petroleum and Minerals, Saudi Arabia. The funders had

no role in study design, data collection and analysis, decision to publish, or preparation of the manuscript.

## Grant Disclosures

The following grant information was disclosed by the authors:

Interdisciplinary Research Centre for Intelligent Secure Systems, King Fahd University of Petroleum and Minerals, Saudi Arabia.

## Competing Interests

The authors declare that they have no competing interests.

## Author Contributions

- Mona Albinali conceived and designed the experiments, performed the experiments, analyzed the data, performed the computation work, prepared figures and/or tables, authored or reviewed drafts of the article, and approved the final draft.
- Mahmood Niazi analyzed the data, authored or reviewed drafts of the article, and approved the final draft.
- Mohammad Alshayeb analyzed the data, authored or reviewed drafts of the article, and approved the final draft.
- Sajjad Mahmood analyzed the data, authored or reviewed drafts of the article, and approved the final draft.
- Arif Ali Khan analyzed the data, authored or reviewed drafts of the article, and approved the final draft.

## Data Availability

This is a literature review.

## Supplemental Information

Supplemental information for this article can be found online at http://dx.doi.org/10.7717/peerj-cs.3005#supplemental-information.

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
