# Peer review of "Taxonomy-based approach for understanding and enhancing security culture in universities"

_PeerJ Computer Science, doi:10.7717/peerj-cs.3005_

## Round 0.1 · original submission · Major Revisions

Dear authors,

For the paper to be considered suitable for publication in PeerJ Computer Science, major changes are required based on the reviewers' comments. Therefore, a significant revision is needed. Consider all the improvements and changes suggested by the reviewers.

We hope you will consider these changes and return your paper.

Best regards,

Emilia Cambronero
Academic Editor for PeerJ Computer Science

Reviewer 1 ·

Basic reporting

1. Is the review of broad and cross-disciplinary interest and within the scope of the journal?

This review offers knowledge that could contribute meaningfully to the literature on Information Security Culture (ISC), especially by organizing and synthesizing findings from existing research. However, several areas require improvement for the paper to fully meet the journal’s standards:

Contribution to Current Findings: The authors should clarify how their review complements or extends current findings, particularly as some studies have already explored the topics they address. This explanation would help demonstrate the unique value and purpose of this review.

Introduction and Terminology: The Introduction section requires rewriting to clearly define ISC and Cybersecurity Culture (CSC) and provide readers with a clear direction for understanding the scope and purpose of the work. Addressing these terms early on will set a stronger foundation for the paper.

Methodology and Article Selection: The Methodology section should provide specific examples of article selection criteria. Since the focus is on ISC within the university sector, the authors should clarify whether the articles they selected specifically discuss ISC in this context. Based on the search strings used, it is unclear whether they would yield ISC-focused articles in academic settings. Additionally, some issues arose when testing these search strings in Web of Science (WoS) databases, suggesting the methodology may need refining to ensure accuracy and relevance.

2. Has the field been reviewed recently? If so, is there a good reason for this review (different point of view, accessible to a different audience, etc.)?

My search in the Web of Science database revealed that several recent reviews have been published in the ISC area. Therefore, to justify the need for this paper, the authors should include a comparative analysis that highlights how their work complements or contrasts with existing reviews. Furthermore, the use of ISC and CSC as central terms needs further clarification, as it influences the literature selection process.

Several high-impact studies, such as those by da Veiga et al. (2020), Tolah et al. (2021), and Nasir et al. (2019), have addressed ISC concepts in detail. If the authors aim to present a new objective (such as in Section 5.1 on "Security Culture Knowledge Area"), they should clarify how their objectives and approach differ from or add to existing literature.

3. Does the Introduction adequately introduce the subject and make it clear who the audience is/what the motivation is?

The Introduction requires revisions to better introduce the subject matter, outline the key issues, describe the current state of research in the field, and specify the objectives the review aims to achieve.

ISC vs CSC Terminology: The authors need to decide whether ISC or CSC is the primary subject of the paper, as the use of these terms is inconsistent throughout. The introduction of both terms without clear distinctions may confuse readers, especially since both ISC and CSC are referenced in the title, abstract, and various sections of the paper. For instance, in the abstract (lines 49 and 58) and body (lines 141 and 142), ISC and CSC seem to be used interchangeably.

If the authors intend for ISC and CSC to be understood as equivalent, they should provide a clear justification, backed by citations. Alternatively, if there are distinct differences, these should be explained in a way that guides readers on the rationale for using both terms. Given the focus on taxonomy, a classification of ISC and CSC would add depth and clarity. Additionally, it may be beneficial to revise the title to reflect this focus.
In summary, while the review has potential to contribute to ISC literature, revisions are needed to clarify its purpose, enhance methodological rigor, and improve terminological consistency.

Experimental design

1. Is the Survey Methodology consistent with a comprehensive, unbiased coverage of the subject? If not, what is missing?

The methodology used in the paper appears suitable for achieving the stated objectives. However, several areas require further elaboration and justification, as the methodology is crucial to ensuring the quality and reliability of the study’s findings. The following points should be addressed to enhance clarity and rigor:

Tables 4 and 5 (Quality Criteria for GL and FL): These tables, which are intended to assess the quality criteria for Grey Literature (GL) and Formal Literature (FL), need additional explanation. Readers need to understand the specific criteria used and how they contribute to selecting papers. For example, clarifying the selection process in Table 5 would be particularly beneficial. I recommend expanding on this in subsection 4.2.4 to improve reader comprehension of the quality assessment.

Final List of Sources: It would be helpful to include a complete list of the final 27 GL and 73 FL sources used in the study, as this will support transparency and allow for a more thorough evaluation of the selected literature.

2. Are sources adequately cited? Quoted or paraphrased as appropriate?

Most sources in the paper are cited correctly, but there are areas needing attention:

Motorola Assessment: The paper refers to the "Motorola Assessment" without any citation or further explanation. Providing a source or a brief explanation would be beneficial for readers unfamiliar with this assessment.

3. Is the review organized logically into coherent paragraphs/subsections?

The organization of the review would benefit from additional visual support:

Process Diagram: Including a flowchart or diagram to illustrate the entire review process—from searching GL and FL sources to deriving findings based on the objectives—would help clarify the methodology. At times, the structure can be challenging to follow, so a visual representation would aid readability and understanding.

Validity of the findings

1. Is there a well-developed and supported argument that meets the goals set out in the Introduction?

The paper references a number of high-impact journal articles that have already contributed to the Information Security Culture (ISC) knowledge base, such as works by da Veiga et al. (2020) and Tolah et al. (2021). These articles cover core concepts within ISC, which overlap with aspects outlined in this paper, particularly in sections related to Research Question 1 (RQ1) and Figure 7. Several areas within Figure 7 appear to reflect themes previously explored in other studies, and some seem similar to operational definitions used in prior ISC research, such as by Nasir et al. (2019).

To thoroughly assess the coverage of the topic, I would need access to the final list of 73 formal literature sources (FL) to confirm that all articles indeed pertain to ISC and are relevant to this paper’s objectives. Without this assurance, the findings—especially the taxonomy objective focusing on ISC in universities—could be compromised.

Additionally, I recommend enhancing the clarity of Figures 8, 9, and 10 to improve readability; ensuring these figures are sharp and well-defined would aid in communicating key findings effectively.

One of the most compelling aspects of this study is found in Appendix B. However, further explanation is needed on the numbering system used in Appendix B, how it relates to the ISC knowledge areas, and the significance of the color coding (blue and black). Clarifying these elements will help readers fully appreciate the value of these findings.

2. Does the Conclusion identify unresolved questions, gaps, or future directions?

The conclusion appropriately addresses the study’s objectives based on the findings presented. However, before finalizing, the authors should address the comments noted above to ensure that the conclusion effectively answers the identified research gaps. This will provide a stronger foundation for the suggested directions and implications highlighted in the conclusion.

Additional comments

References
da Veiga, A., Astakhova, L. V., Botha, A., & Herselman, M. (2020). Defining organisational information security culture—Perspectives from academia and industry. Computers and Security, 92, 101713. https://doi.org/10.1016/j.cose.2020.101713
Nasir, A., Abdullah Arshah, R., & Ab Hamid, M. R. (2019). A dimension-based information security culture model and its relationship with employees’ security behavior: A case study in Malaysian higher educational institutions. Information Security Journal, 28(3), 55–80. https://doi.org/10.1080/19393555.2019.1643956
Tolah, A., Furnell, S. M., & Papadaki, M. (2021). An empirical analysis of the information security culture key factors framework. COMPUTERS & SECURITY, 108. https://doi.org/10.1016/j.cose.2021.102354

Additional comments
1. Please check the format. Some subsections are not numbered. Please check the sequence number properly. Some are mixing with capital letters. For example in line 551.
2. You should write something in subsection rather than leave it blank in line 541

Reviewer 2 ·

Basic reporting

The primary focus of this study is to examine and define the concept of cybersecurity culture within universities, particularly in the context of Saudi Arabia. It seeks to address gaps in existing literature by developing a taxonomy of best practices that can enhance information security culture (ISC) across higher education institutions (HEIs). By exploring the unique cybersecurity challenges faced by universities and recognizing socio-cultural factors that impact security behaviors, this study aims to provide actionable insights for fostering a proactive, secure environment within educational settings, ultimately contributing to a framework that prioritizes cybersecurity in academia.

However, to improve the quality of the manuscript, please find below some of my comments.

Assessment #1:
The article mostly used a clear and professional English suitable for an academic context.

Comment #1:
However, better to check and improve the sentence structures throughout the study.

Assessment #2:
The introduction is well-structured, providing a logical flow from general information security issues to the specific focus on Information Security Culture (ISC). Also, the paper clearly introduces the importance of information security, the need for a culture of security awareness, and how it applies to universities, especially in Saudi Arabia.

Comment #2:
The introduction could benefit from a more targeted discussion on the unique information security challenges faced by universities, such as students, academic staff, and non-academic staff data protection, academic collaborations, and the open nature of university networks. A clearer distinction between general organizational issues and those specific to educational institutions would strengthen the narrative.

In addition, while the introduction mentions the importance of developing a culture of security awareness, it doesn’t address the socio-cultural factors that influence how different groups within universities (students, academic staff, non-academic staff) approach security. Different regions may have different attitudes towards technology and security.

Comment #3:
The introduction could include a brief mention of socio-cultural differences in security attitudes within universities, especially in Saudi Arabia, where digital literacy and cultural norms might influence the effectiveness of security awareness programs.

Assessment #3:
The literature review provides a broad and varied look at cybersecurity culture, encompassing various regions (e.g., Saudi Arabia, the Middle East, and India) and levels (such as universities and the general public). This adds depth to the discussion and presents a well-rounded picture of cybersecurity challenges in educational institutions.

Comment #4:
The structure of the literature review could be improved. It jumps between discussing different countries and specific findings without a clear transition. Grouping studies by themes (e.g., awareness levels, policy issues, behavioral aspects) rather than by region could enhance readability and thematic coherence.

Experimental design

Assessment #4:
The methodology section is significantly longer than other sections in the manuscript.

Comment #5
To improve clarity, it would be beneficial to include a figure that visually represents the methodology in a step-by-step format for the entire study. This figure should outline the total number of phases involved, as well as any steps or sub-steps under each phase, providing a clear indication of the methods and processes used.

E.g., in your case, "Phase 1 (Planning the MLR)" and "Phase 2 (conducting the review)", mention these first and then write sub-steps under these phases. This visual aid will help readers better understand the study's approach and follow the progression of the research more easily.

Validity of the findings

Assessment #5:
The discussion covers various knowledge areas that are essential for establishing a cybersecurity culture in universities, such as top management support, effective security policies, awareness and training, compliance, risk assessment, communication, trust, and ethical behavior. This offers a good view and ensures that key components of cybersecurity culture are addressed.

Comment #6:
The discussion could benefit from more extensive citation of recent and relevant studies in the context of HEIs. Most of the statements are presented without direct reference to prior research, which might undermine the academic rigor of the argument.

Comment #7:
Organize the discussion by grouping similar concepts and streamlining redundant explanations. E.g., security policies, compliance, and ethical behavior could be discussed under a single section on policy-driven security culture, with subheadings to delineate specific points.

Additional comments

General comments:
The review could benefit from more extensive citation of recent (2024 if there is any) and relevant studies in the context of HEIs.
Most of the statements are presented without direct reference to prior research (in the discussion section), which might undermine the academic rigor of the argument.

Also, texts in both the Figures 8 and 9 are difficult to read. Please try to make it more readable.

·

Basic reporting

While the conclusion identifies gaps in current knowledge, it lacks specific and actionable suggestions for future research. For instance, the call for a "security culture readiness model" is underexplored and would benefit from a clear outline of the steps needed for development.
The methodological explanation lacks precise detail, particularly in the data extraction phase of the Multivocal Literature Review (MLR). The description of the selection criteria and inclusion of grey literature sources is not sufficiently detailed, raising concerns about potential selection bias

Experimental design

the study design demonstrates a structured approach to reviewing literature on security culture but could benefit from improved transparency, methodological rigor, and a more systematic approach to grey literature inclusion.

Validity of the findings

The research findings in this article identify key knowledge areas and best practices for enhancing information security culture in universities, with the study resulting in a taxonomy aimed at organizing these insights.

Additional comments

Under each knowledge area, the study lists best practices, providing actionable insights for universities. This level of detail is beneficial as it offers specific, contextually relevant strategies, making it easier for university administrators to apply the findings

---

## Round 0.2 · Major Revisions

Dear Authors,

Thank you for submitting your manuscript to PeerJ Computer Science. After a thorough review, we have determined that significant revisions are necessary for the paper to be considered suitable for publication. The reviewers have provided detailed comments and suggestions for improvement, which we encourage you to address comprehensively.

We look forward to receiving your revised manuscript and appreciate your efforts to make the necessary changes.

Best regards,

M. Emilia Cambronero
Academic Editor
PeerJ Computer Science

Reviewer 1 ·

Basic reporting

1.Comparative Analysis and Scope of the Review

The inclusion of comparative analysis tables (Table 1 and Table 2) is a positive addition; however, the comparison should be limited to studies focusing on educational settings. Including studies from other organizational settings does not help in narrowing the scope or reinforcing the justification for this paper.

2. Terminology and Consistency

The decision to use only ISC instead of ISC and CSC has improved clarity. However, further refinements may be necessary (see comments below regarding methodology and findings).

Experimental design

1. Methodology and Article Selection

The primary objective of this study is to examine ISC in university settings and develop a taxonomy specifically for universities. However, several of the included papers do not focus on ISC within academic environments. This is a fundamental issue, as it affects the reliability and validity of the proposed taxonomy.

If the authors choose to include papers from non-university settings, they must provide a strong justification and explain:

Why these papers were included.

Why papers that did not meet 50% of the inclusion criteria were still considered relevant.

How these papers contribute to ISC in universities, specifically.

The authors have correctly stated that they applied exclusion criteria (Table 5) to filter out papers that were not relevant to the research question. Given this, it is essential that one of the exclusion criteria explicitly removes papers that do not focus on ISC in university settings. This is a critical issue because the title, research questions, and objectives of this study clearly center on ISC in universities. Including studies from non-university environments undermines the reliability of the findings and the validity of the proposed taxonomy. I strongly urge the authors to refine their selection criteria to ensure alignment with the study’s stated scope.

The search strings seem to capture ISC-related studies broadly, but without an appropriate exclusion mechanism, the results do not align with the study’s stated objectives.

Validity of the findings

The taxonomy developed is based on papers that include non-university settings, which weakens the study's central claim. If the taxonomy aims to reflect ISC exclusively in universities, then the sources must be strictly relevant to that context.

Additional comments

1. Formatting and Writing Discipline

Several formatting errors remain throughout the document, including:

Figure 6 (Capitalization issue in "Information Security").

Font inconsistencies (e.g., Line 366, Figure 1 font differs from other figures; similar issues at Line 350, 386).

In-text citation formatting (e.g., space inconsistencies between text and citation in Lines 359, 366, 382).

2. Misleading Statements

The phrase "bridge the gap between academia and the software industry" is misleading. ISC pertains to organizational security culture, not software quality. This should be revised to reflect academia and organizational security practices.

3. Ethical Concerns: AI-Generated Image in Figure 2

The AI-generated image in Figure 2 is unacceptable for academic publication. It contains spelling errors, misleading arrows, and incorrect representations that do not accurately convey the thematic analysis process. The diagram does not enhance understanding and, in fact, detracts from the paper’s credibility.

Recommendation: Replace Figure 2 with a properly structured diagram that accurately represents thematic analysis, ensuring clarity and correctness.

4. Conclusion and Final Evaluation

The Conclusion section has been revised to address unresolved questions and research gaps. However, this section should also justify the methodology used, reinforcing why the selection and analysis of studies are appropriate for the study’s objectives.

Final Decision: Major Revisions Still Required

While the authors have significantly improved the paper, the methodology and findings remain problematic due to the inclusion of non-university-related ISC papers in the analysis.

Formatting and clarity must be improved, and the AI-generated figure should be removed or replaced with a more accurate visual representation.

If the authors insist on including non-university-related ISC papers, a strong justification must be provided. Otherwise, the paper cannot be considered for publication in its current form.

I appreciate the authors' hard work in addressing my previous comments. However, I strongly recommend that they carefully refine their methodology, findings, and formatting to ensure that their research meets the standards required for high-impact journal publication.

Reviewer 2 ·

Basic reporting

I have reviewed the revised manuscript, and all my concerns have been addressed. I have no further suggestions at this stage.

Experimental design

No Suggestion

Validity of the findings

No Suggestion

Additional comments

No Suggestion

---

## Round 0.3 · Minor Revisions

Dear Dr. Albinali,

Since its last revision, the article has improved, but its format needs to be presented more carefully. Please, make sure that all pages are free of blank spaces, that all figures respect the margins, and that the same font and font size are used throughout, including the text of the figures. The paper format must be improved to be able to be published in PeerJ Computer Science.

Best regards,

M. Emilia

---

## Round 0.4 · accepted · Accept

Dear Authors,

I am pleased to inform you that your article has been accepted for publication in PeerJ Computer Science.

Best regards,

Emilia Cambronero
Academic Editor for PeerJ Computer Science